# Fatigue Performance of Type I Fibre Bragg Grating Strain Sensors

**DOI:** 10.3390/s19163524

**Published:** 2019-08-12

**Authors:** Naizhong Zhang, Claire Davis, Wing K. Chiu, Tommy Boilard, Martin Bernier

**Affiliations:** 1Department of Mechanical and Aerospace Engineering, Monash University, Wellington Rd, Clayton, VIC 3800, Australia; 2Defence Science and Technology Group, 506 Lorimer Street, Fishermans Bend, VIC 3207, Australia; 3Centre d’Optique, Photonique et Laser (COPL), Université Laval, Québec, QC G1V 0A6, Canada

**Keywords:** fibre Bragg gratings, grating inscription method, fibre fatigue test rig, FBGs fatigue testing, optical properties

## Abstract

Although fibre Bragg gratings (FBGs) offer obvious potential for use in high-density, high-strain sensing applications, the adoption of this technology in the historically conservative aerospace industry has been slow. There are several contributing factors, one of which is variability in the reported performance of these sensors in harsh and fatigue prone environments. This paper reports on a comparative evaluation of the fatigue performance of FBG sensors written according to the same specifications using three different grating manufacturing processes: sensors written in stripped and re-coated fibres, sensors written during the fibre draw process and sensors written through fibre coating. Fatigue cycling of the fibres is provided by a customized electro-dynamically actuated loading assembly designed to provide high frequency and amplitude loading. Pre- and post-fatigue microscopic analysis and high-resolution transmission and reflection spectra scanning are conducted to investigate the fatigue performance of FBGs, the failure regions of fibres as well as any fatigue-related effects on the spectral profiles. It was found that because of the unique fabrication method, the sensors written through the fibre coating, also known as trans-jacket FBGs, show better fatigue performance than stripped and re-coated FBGs with greater control possible to tailor the optical reflection properties compared to gratings written in the draw tower. This emerging method for inscription of Type I gratings opens up the potential for mass production of higher reflectivity, apodised sensors with dense or complex array architectures which can be adopted as sensors for harsh environments such as in defence and aerospace industries.

## 1. Introduction

Fibre Bragg gratings (FBGs) have become one of the major elements in various fibre-optic devices over the past decades [1]. FBGs are periodic variations in the refractive index of the core of the fibre. They are inscribed by first photosensitizing the glass and then exposing the fibre core side on to laser light with a spatially modulated intensity pattern [2,3]. The grating is designed to act as a narrow band reflector, reflecting light of a specific wavelength known as the Bragg wavelength and transmitting light at all other wavelengths. The Bragg wavelength is expressed as
(1)$$\lambda_{B}=2n_{eff}\Lambda$$
where *λ_B_* is the Bragg wavelength, *n_eff_* is the effective refractive index of the FBG and *Λ* is the modulation period of the index change. The Bragg wavelength is directly dependent on both the strain and temperature as shown in Equation (2),
(2)$$\frac{∆\lambda_{B}}{\lambda_{B}}=\left(\alpha +\xi \right)∆T+\left(1-\rho_{e}\right)\varepsilon$$
where ∆*T* is the temperature change experienced at the FBG sensor location, *α* is the thermal expansion, *ξ* is the thermo-optic coefficient, *ε* is the longitudinal strain on the FBG and *ρ_e_* is the effective photo-elastic constant of the fibre core material [4]. Hence strain and/or temperature can be measured by monitoring the reflected Bragg wavelength and the strain and temperature contributions can be de-coupled from one another by using a variety of techniques [5,6]. Most FBG sensors are designed to suit interrogation devices which are widely available at low cost in the telecommunications C-Band wavelength range (1530–1565 nm) with a typical strain sensitivity of approximately 1.2 pm/microstrain (µε) and 10 pm/°C [7].

FBGs have found widespread use in structural health monitoring (SHM) applications particularly with the rapid development of glass fibre photosensitization techniques and the evolution of new FBG sensor fabrication methods. FBGs are recognized as having many advantages compared with conventional electrical resistance foil gauges due to their corrosion resistance, immunity to electromagnetic interference (EMI), light weight and small physical dimensions facilitating embedded deployment of FBGs into structures for long-term strain monitoring purposes [8].

However, the reported reliability and durability of optical fibres containing FBGs sensors has been variable, in part because of the fabrication methodologies used to create the sensors [9]. The fabrication processes for FBGs can be categorized into three main types: coating stripped and recoated gratings, draw tower gratings, and through-coating gratings also known as trans-jacket gratings.

The first FBG fabrication method removes the protective coating from the glass in the region of the fibre where the FBG is to be inscribed. The rationale for employing this process is that many of the UV laser sources typically used to inscribe the grating do not transmit effectively through the fibre coating. The fibre is also photosensitised, this may occur either before or after the coating removal process depending on whether heat is applied as part of the process. Finally, the grating is thermally annealed and recoated post-inscription. Each one of these processes involves manual handling and can introduce surface flaws in the unprotected region of the glass which can in turn affect its long-term reliability and durability [10]. It has been previously reported that the mechanically stripped and recoated FBGs have decreased tensile strength, compared with their pristine counterparts [11]. Ang et al. [12] reported a peak fatigue strain of 5000 µε when a stripped and re-coated FBG embedded in a composite material was tested which is only 10% of the tensile strain limit for pristine fibres.

Another way to write the gratings is in the fibre drawing tower during the fibre fabrication process, these gratings are referred to as draw tower gratings. Two interfering beams of the UV laser light write the Bragg gratings after the formation of the fibres prior to the application of the protective coating. This method of FBG fabrication has several advantages: The grating can be accurately positioned at its desired location; the process induces no mechanical damage on the fibre coating and arrays of gratings can be inscribed at fixed intervals using an automated process. The FBG sensors manufactured by this method are expected to have good mechanical performance under harsh environmental conditions. However, the time available to expose the fibre is limited to a single pulse by the fibre draw speed and hence the maximum reflectivity of the grating which can be achieved is limited to a few percent. In addition, there is no opportunity to apodize the index profile of the grating meaning that grating side lobes cannot be suppressed. Both of these factors limit the quality of the resultant reflection spectra [13].

Many of the polymers commonly used as coatings for protection of the fibre are transparent in the IR region of the spectrum, which opens up the opportunity for using IR lasers to write gratings into the core of the optical fibre through the coating. The ability to write through the coating has simplified the FBGs fibre fabrication process dramatically and significantly reduced the manufacturing overheads. Infrared (IR) femtosecond pulse duration (ultrafast) radiation can penetrate and induce changes in the refractive index of any material that is transparent to low-signal-intensity IR radiation without the requirement to have high dopant concentrations in the glass. It is not necessary to strip off the fibre coatings prior to the laser exposure and to recoat the gratings afterwards, which previous research has shown to introduce limitations on the fibre robustness [14]. The peak reflectivity achievable using such infrared femtosecond inscription processes is significantly higher compared with draw tower inscription process and the resulting gratings have been reported to maintain the mechanical strength of the pristine fibre, with the capability of achieving robust distributed FBG arrays made by strain tuning [15,16]. To date though, no studies have been reported on the fatigue performance of FBGs written using this method.

This paper presents data on a comparative evaluation of the overall fatigue performance and optical properties of a total of 45 unpackaged FBGs, 15 fabricated using each of the three processes. The gratings were subjected to tensile fatigue loading with their optical reflection and transmission spectra characterised pre- and post-fatigue with the aim of establishing whether trans-jacket gratings FBGs exhibit performance improvements in both mechanical and optical behaviour.

The driver for this study was to investigate the potential for FBGs sensors to replace conventional electrical resistance strain gauges on high value defence platforms operating in harsh environments. The ability to multiplex large numbers of sensors into a single fibre and significantly reduce the scale and complexity of the wiring is also seen as a major benefit for these applications. In a recent paper, it was reported [17] that the robustness of electrical resistance strain gauges are often unsuitable for long-term full-scale fatigue testing (FSFTs) of military platforms. The complex wiring associated with the large number of gauges required can lead to excessive weight on the structure under test which can influence the results. This paper aims to provide experimental evidence about the fatigue response and long-term mechanical performance of FBG sensors.

## 2. Experimental Methods

The fatigue study was conducted in four stages; design, characterisation and optimisation of the optical fibre fatigue loading assembly; pre-fatigue microscopic and spectral analysis of the sensors; fatigue loading of the sensors and post-fatigue microscopic and spectral analysis of the sensors. Each stage is described in more detail in the proceeding sections.

### 2.1. Optical Fibre Fatigue Loading Assembly

Conventional servo-hydraulic mechanical testing systems are typically confined to low cycle fatigue rates which limits the practicality of their use for large volume, high cycle number testing applications. In addition, their loading capacity is typically excessive for fatigue cycling of samples with small cross-sectional areas such as optical fibres, which will require operation within the lowest 5% of the range where the hydraulic actuators have difficulty maintaining effective and accurate control.

Therefore, a new electro-dynamically actuated high frequency loading assembly was designed and constructed to address the engineering constraints associated with servo-hydraulic loading. The dynamic range of the system was designed to achieve tensile strains in the optical fibre of up to 25,000 µε at cyclic loading rates of up to 100 Hz. The fatigue test was designed to apply constant amplitude sinusoidal loading and relied on actuation authority provided by a high capacity electrodynamic shaker (TIRA GmbH, S50350, Schalkau, Germany) controlled by a Vibration Research Corporation 8500 vibration controller. The fibre was mounted between two blocks comprising the mechanical grips, one fixed and one free to move with pre-tensioning of the fibres applied by a screw-driven actuator and a single spring to facilitate the dynamics of the movable block, as shown in Figure 1.

The dynamic configuration of the test rig mainly comprises a tensile spring that connects the moving part and the top side of the assembly. The fibre which is being tested is mounted between the moving part and the fixed base of the assembly. The pre-tension of the fibre is achieved via the extension of the tensile spring with stiffness K_1_. When the base is excited by the shaker, the dynamic response of the moving part will subject the fibre (denoted by stiffness K_2_) to an oscillatory fatigue loading. The simplified dynamic system is shown in Figure 2.

The fibre strain is applied via the relative motion of the base and the moving block. Essential parameters that influence the vibration characteristics of the test rig are the tensile spring stiffness $K_{1}$, fibre stiffness $K_{2}$, and the mass of the moving part M. The sinusoidal excitation from the shaker is represented by an input displacement $X_{i}\left(t\right)=X_{0}\sin\left(\omega t\right)$. The displacement of the moving block is expressed as the signal X(t). Therefore the fibre strain is expressed in terms of the relative displacement between $X_{i}\left(t\right)$ and X(t). Neglecting the effects of damping, the equation of motion for mass M [18] is expressed as Equation (3),
(3)$$M\ddot{x}+\left(K_{1}+K_{2}\right)\left(x-x_{0}\right)=0$$
assuming the moving part oscillates at the same frequency ω:(4)x=Xsin(ωt)

The displacement ratio of the moving part and the shaker base is:(5)$$\left|\frac{X}{X_{0}}\right|=\frac{1}{1-\left(\frac{\omega }{\omega_{n}}\right)}$$
where ω is the shaker operating frequency, and the natural frequency of the system, $\omega_{n}=\sqrt{\frac{K_{1}+K_{2}}{M}}$. The relative displacement of moving block, and hence the fibre strain, can be maximized when the mass is operated at frequencies close to the natural frequency of the test rig. The parameters for the loading assembly used are shown in Table 1:

The stability of the test fixture was achieved by increasing the input frequency slightly above resonance. A reasonable balance between testing speed and strain amplitude was achieved with an input frequency of 100 Hz. Under these conditions, strain levels in the fibre of up to 25,000 µε could be reliably maintained for a large number of load cycles.

The finalised optical fibre loading assembly is shown in Figure 3. It was bolted to a platform installed on the shaker, where the vibration was supplied. The wedge-shaped jaws were designed to grip the fibre without slipping even at very high strain levels during fatigue testing. Rubber inserts were used with the jaws to protect the fibre from the clamping pressure. An adjustable tensile force was applied by a screw actuator to pre-tension the fibre such that the neutral strain level could be set to the mid-point of the total excursion. By using an interrogator, one can monitor the wavelength as well as the transformed strain value using the relation of 1.2 pm/microstrain.

FBG sensors are sensitive to both temperature and stress variations. The sourced parameter from suppliers for a standard FBGs has a temperature sensitivity of ~10 pm/°C [19]. The temperature in the laboratory in which fatigue experiments are conducted is 23 °C ± 5 °C. This variation in temperature is expected to result in a wavelength shift in a range of ±50 pm. Therefore, the resultant strain error expected is ±0.16% of 25,000 µε, which is negligible compared with the strain levels used in the fatigue experiments.

### 2.2. Microscopic Analysis of the Fibre Surface

Before the fatigue testing, microscopic inspection was conducted at the FBG sensor location for each of the fibres, to examine for any physical damage on the fibre surface. Stripped and recoated FBGs have previously been reported [20] with surface flaws caused by the stripping and/or recoating method. After the fatigue experiments, all the fibres were examined again under a microscope to confirm their failure location (if failed) and to inspect for any fatigue damage of the fibre or fibre coating in the unfailed fibres. These inspections, combined with the fatigue data from the experiments, were conducted to provide evidence to see if the fabrication methods can influence fatigue properties of FBGs.

### 2.3. Spectral Analysis of FBG Reflection and Transmission Properties

The optical reflection and transmission properties of FBG sensors are mainly determined by grating length, index contrast and grating pitch which all contribute to determine the reflection and transmission profile [21]. A 1pm resolution tunable laser (Yenista, TS100, Lannion, France) combined with a component tester (Yenista, CT400, Lannion, France) characterized the reflection and transmission spectra of the FBGs before and after their fatigue testing. These scans allow the key features in the FBG spectra associated with each manufacturing method to be characterized. The high optical power and wavelength resolution also allows accurate determination of any changes to the spectral profile after fatigue that can indicate material or coating damage.

### 2.4. FBG Tensile Fatigue Experimental Setup

The complete experimental arrangement for conducting the fatigue loading of the optical fibres is shown in Figure 4. The load was applied using a high capacity electrodynamic shaker (TIRA GmbH, S50350, Schalkau, Germany) driven by a vibration controller with a sinusoidal input at approximately 100 Hz (see Section 2.1). The amplitude of the controller could be adjusted to achieve a strain range of up to 25,000 µε. The resultant strain induced in the FBG sensor was measured continuously using an industrial grade optical sensing interrogator (Micron Optics, Si255, Atlanta, GA, USA). This interrogator features a dynamic strain range of approximately 130,000 µε across a wavelength range from 1460–1620 nm. The unstrained centre Bragg wavelength for all of the FBGs tested was between 1530 and 1550 nm, meaning the anticipated strain was well within the range of the interrogator.

A series of 45 individual type I FBG sensors as detailed in Table 2 were tested according to an incremental fatigue loading schedule as outlined in Table 3. All the FBG sensors were provided by commercial suppliers with the exception of 10 of the trans-jacket gratings which were provided by an academic researcher under contract. All the fibres tested had the same physical dimensions with a fibre diameter of 125 µm and a polyimide (PI) coating of 15 µm with the exception of the draw tower gratings which had an Ormocer^®^ coating of the same thickness. All of the sensors were ordered with the same gauge length (5 mm) and fundamental Bragg centre wavelength (in the range of 1530 nm to 1550 nm). The draw tower gratings were written using an interferometric process with a pulsed UV laser. The stripped and re-coated gratings has the coating removed via a chemical etching process with the grating inscribed with a UV laser using phase mask inscription. The trans-jacket gratings were written with femtosecond IR laser pulses at 800 nm using phase mask inscription. The fibres were tested until failure or completion of the loading schedule, whichever occurred first.

The total number of loading cycles accumulated to 4 million which took about 11.2 h for a single FBG sensor to complete the experiment. The fatigue performance of the sensors was better than initially expected so the number of cycles at the higher strain levels was reduced to 0.5 M cycles in order to maintain a reasonable test duration for each sensor.

## 3. Results and Discussion

### 3.1. FBG Fatigue Performance

Figure 5 summarises the main results from the fatigue testing. Only two of the fifteen stripped and re-coated FBGs survived the entire fatigue loading schedule compared with five of the fifteen draw tower gratings and two out of the five commercially supplied trans-jacket gratings. All of the trans-jacket gratings provided by the research provider survived the entire test schedule.

The bar chart illustrates that for the failed fibres the draw tower gratings had the highest mean fatigue failure strain at 23,000 µε, the commercially supplied trans-jacket gratings had a mean fatigue failure strain of 20,750 µε and the stripped and recoated gratings had the lowest mean fatigue failure strain at 18,800 µε. The error bars show the standard deviation of the fatigue failure strain results. The largest variability in fatigue failure strains is for the stripped and re-coated fibres with a standard deviation of nearly 5000 µε. Figure 6 and Figure 7 provide some further detail on the distribution of this data. As expected, for the stripped and recoated gratings, all of the failures occurred within or near the boundaries of the stripped section of the fibre. The large range of failures strains is also unsurprising due to the number of stages and degree of handling involved in fabricating a stripped and re-coated grating and the potential for damage of the exposed glass during any of those stages. The range of fatigue failure strains for the draw tower gratings and the commercially supplied trans-jacket gratings by comparison were much smaller and the failure locations were distributed randomly across the strained section of the fibre with no concentration in sensing region.

In general, although the fatigue performance of the stripped and re-coated gratings were noticeably inferior to the draw tower gratings and the trans-jacket gratings, the overall results were considered against the reported fatigue performance of electrical resistance foil gauges which are currently the industry standard. Figure 8 shows a plot of the reported fatigue life for commercially supplied standard and fatigue resistant electrical gauges. The results reported in this paper indicate that even the stripped and recoated optical gauges exhibit far better fatigue life than the speciality fatigue resistant (M-Series) electrical gauges which can only withstand one million loading cycles at a fatigue strain amplitude of 2500 µε.

The results from the fatigue study also showed that 100% of the trans-jacket FBGs made by the research provider survived the fatigue loading sequence, which is a promising indicator for long-term performances under very harsh environments. These FBGs were fabricated according to the experimental procedure and parameters detailed in [15]. An acylindrical lens of short focal length (f = 8 mm) was used to tightly focus the 800 nm femtosecond writing beam in the centre of the fibre. Such tight focusing geometry with reduced optical aberrations allows two orders of larger intensity magnitude to be reached around the fibre core compared to the surrounding polymer coating, thus making possible trans-jacket inscription of type-I gratings in the fibre core without damaging the coating [23]. Each of the 10 samples was proof tested after inscription at 70% of the fibre maximum strain at failure (i.e., ~3.6 GPa, 70% of ~5.2 GPa) to ensure its mechanical stability. The emergence of commercially available trans-jacket gratings written using femtosecond lasers is relatively recent and it is expected that as commercial providers adapt and refine their processes for volume production that the performance of these sensors will continue to improve to match those fabricated in low volumes in the research environment.

### 3.2. FBG Pre and Post-Fatigue Physical Inspection

Figure 9 shows a series of representative microscopic images from the surface of the fibre in the region of the sensor for each FBG sensor type. The stripped and recoated fibres all showed surface imperfections with inconsistent coating thicknesses in the recoated sections and steps in the boundary region between the original coating and recoated sections as shown in Figure 9a. These types of surface flaws are common for stripped and recoated gratings and are associated with the layer by layer re-coating approach utilised by commercial fibre re-coaters where the coating material is built up in 3–10 µm layers over the base material with a two-stage thermal cure applied after each layer.

In contrast, there was no evidence of coating damage or coating thickness variabilities for the draw tower gratings, as shown in Figure 9b. This is to be expected for this class of grating where the coating is applied after the grating is inscribed to the entire fibre as part of the normal manufacturing process.

For the trans-jacket gratings, if the laser beam is not tightly focused in the core during the inscription process, there is potential for thermal damage to occur through laser induced heating of the fibre coating if the energy density becomes too high in this region. All of the trans-jacket gratings were examined closely across the entire region of the grating and no surface damage or discolouration was observed indicating that no physically observable damage had occurred.

All the fibres were examined after fatigue cycling and again no noticeable changes to the fibre coating surfaces were observed as a result of the cycling.

### 3.3. Characterisation of Reflection and Transmission Properties

Figure 10 shows representative reflection and transmission spectra for each of the grating types. There are specific features associated with the FBG manufacturing process which are evident for each spectrum. The draw tower gratings all have a much lower reflectivity (<5%) which is associated with the fact that the gratings are written during the fibre draw process where there is a very limited time window available to expose the fibre before it is coated and spooled as part of the active drawing process [24]. This means that the maximum index contrast, which can be achieved by laser exposure is low despite the use of highly photosensitive glass materials and a slow fibre draw rate.

For the stripped and re-coated and trans-jacket gratings, the fibre is static and hence the exposure time can be increased to achieve the required index contrast for higher reflectivity. The reflectivity of these FBGs ranged between ~50% and ~80%. The stripped and re-coated gratings also showed much better side-lobe suppression than the trans-jacket gratings as it is typically easier to accurately focus the light in the core of the fibre and control the apodization features of the index contrast with the coating removed [25]. The degree of side-lobe suppression is significant as it has a direct influence on the number of gratings which can be accommodated within a spectrally dense sensor array before interference from side lobes starts to affect the bounding gratings in the array.

The reflection spectra for each grating was also measured post-fatigue for all the gratings. For the failed gratings, the reflection transmission was measured from each side of the broken fibre to determine where the failure had occurred. A failure within the active sensing region was indicated by a reduced reflectivity spectrum on each side of the fibre. A complete reflection spectrum on the right-hand side of the failed fibre indicated that the fibre had failed to the left of the sensor. Conversely, a complete reflection spectrum on the left-hand side of the failed fibre indicated that the fibre had failed to the right of the sensor. The reflection and transmission spectra for the gratings which survived the complete fatigue loading cycle were compared pre- and post-fatigue. Figure 11 shows a representative plot for a stripped and recoated grating pre and post fatigue which indicates that there are no significant changes to the optical transmission properties of the grating resulting from the fatigue. All the measured spectra following fatigue followed this trend.

## 4. Conclusions

This paper reported the fatigue performance of unpackaged fibre Bragg gratings manufactured using three processes which are currently used by commercial suppliers. The results indicate that draw tower gratings and trans-jacket gratings have superior tensile fatigue performance compared to conventional stripped and recoated grating under high strain cyclic loading. This result is unsurprising and consistent with the reduced manual handling and exposure to the environment that these manufacturing methods require. It should be noted, however that the performance of all grating types is still vastly superior to the reported fatigue performance of electrical resistance strain gauges.

The fatigue performance of the commercially supplied draw tower gratings and trans-jacket gratings were reasonably similar whereas the trans-jacket gratings written in the research environment under a controlled exposure procedure and proof-tested after inscription survived the complete fatigue testing regime of 10 M cycles at strain levels up to 25,000 µε. This is well in excess of anything which might be expected in even the harshest of fatigue loading environments for structural health monitoring. As commercial suppliers continue to adopt and refine their trans-jacket inscription processes for volume production, better performance and lower costs are anticipated. 

In terms of optical performance, the stripped and recoated gratings offer the best potential for optimisation of the spectral properties including side-lobe suppression; however, this may not be necessary for most of the peak tracking algorithms which are used with commercially available FBG interrogators for strain and temperature measurements as long as the peak reflectivity is high enough above the noise floor of the measurement device.

Although mechanically robust, the draw tower grating fabrication process imposes limitations on grating reflectivity, gauge length and sensor density. The trans-jacket inscription process opens up the potential for robust, fatigue resistant sensing higher reflectivity, apodised gratings with denser or more complex array of architectures which can be mass produced at low costs. It is expected that these developments will lead to an increased market penetration for FBG based strain sensors in structural health monitoring.

## Figures and Tables

**Figure 1 sensors-19-03524-f001:**
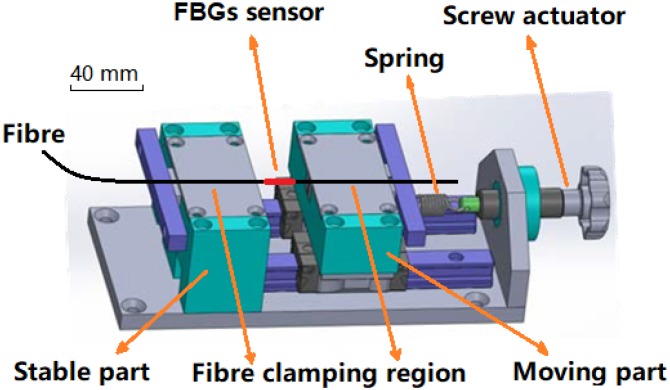
Initial loading assembly design.

**Figure 2 sensors-19-03524-f002:**
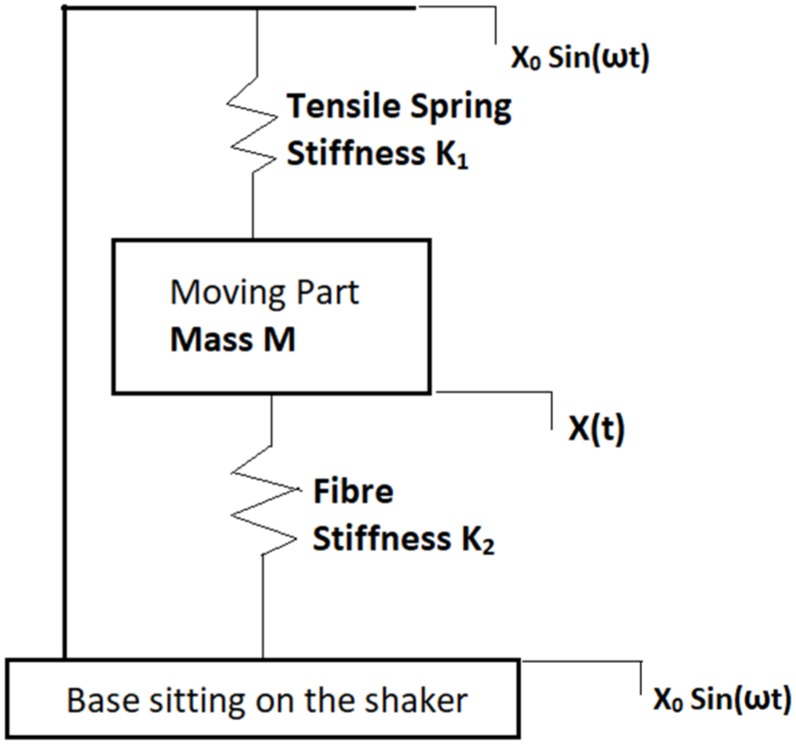
Dynamic configuration of the fibre test rig.

**Figure 3 sensors-19-03524-f003:**
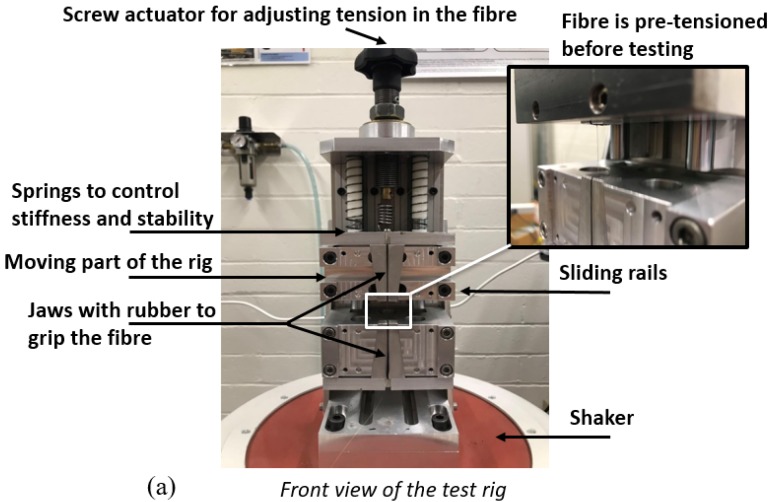
(**a**) Front view of the test rig sitting on the shaker, the fibre Bragg gratings (FBG) is mounted between the moving part and the stable part. (**b**) A functional diagram illustrating working principle of the test rig.

**Figure 4 sensors-19-03524-f004:**
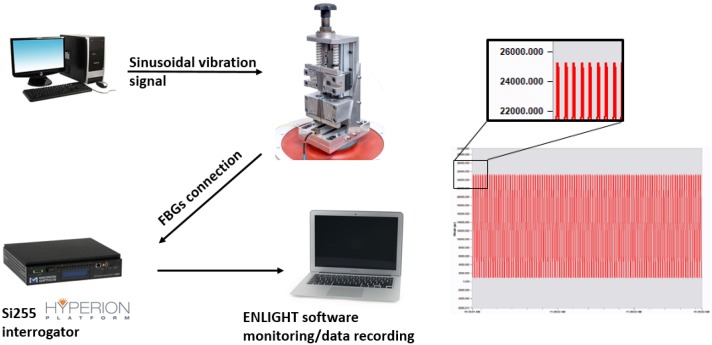
Complete fatigue testing experimental setup including an example time history of the strain response. The figure inset shows a magnification of the peak strain of 25,000 µε at 100 Hz.

**Figure 5 sensors-19-03524-f005:**
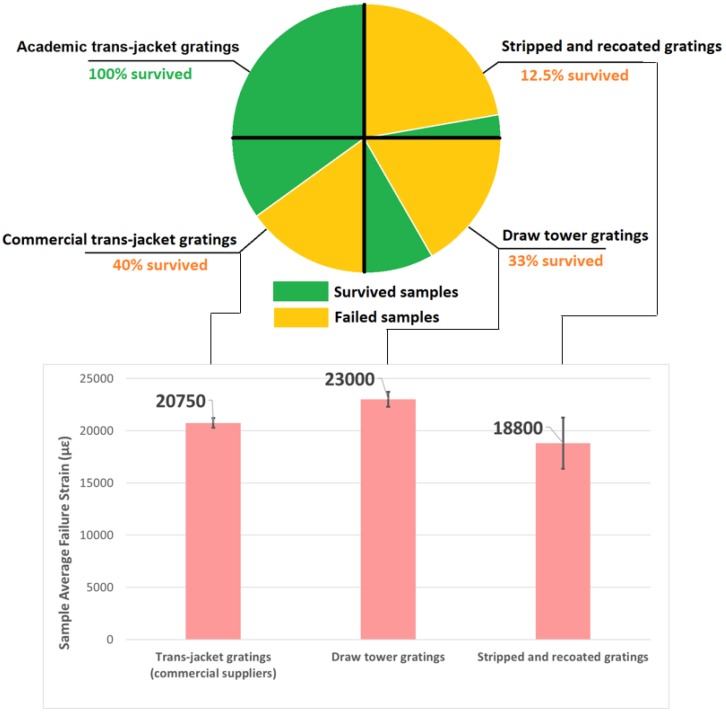
Summary of fatigue performance of FBG sensors, including average strain to failure and survival rates (error bars show the standard deviation of the failure strain results).

**Figure 6 sensors-19-03524-f006:**
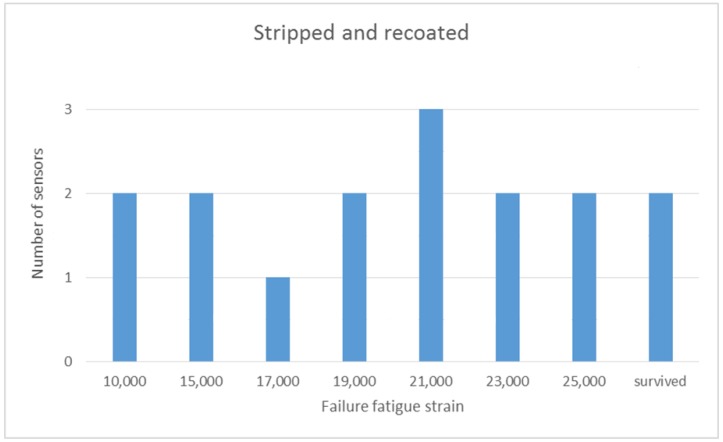
Range of fatigue failure strains for stripped and recoated FBGs.

**Figure 7 sensors-19-03524-f007:**
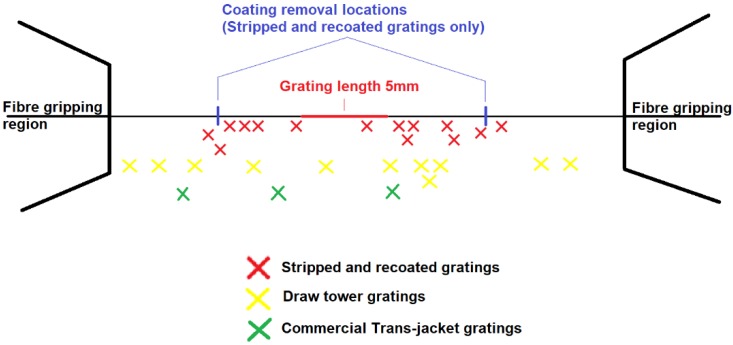
Fatigue failure points marked at their relative positions for different FBGs, to be noticed that there are coating strip-off points on stripped and recoated FBGs only.

**Figure 8 sensors-19-03524-f008:**
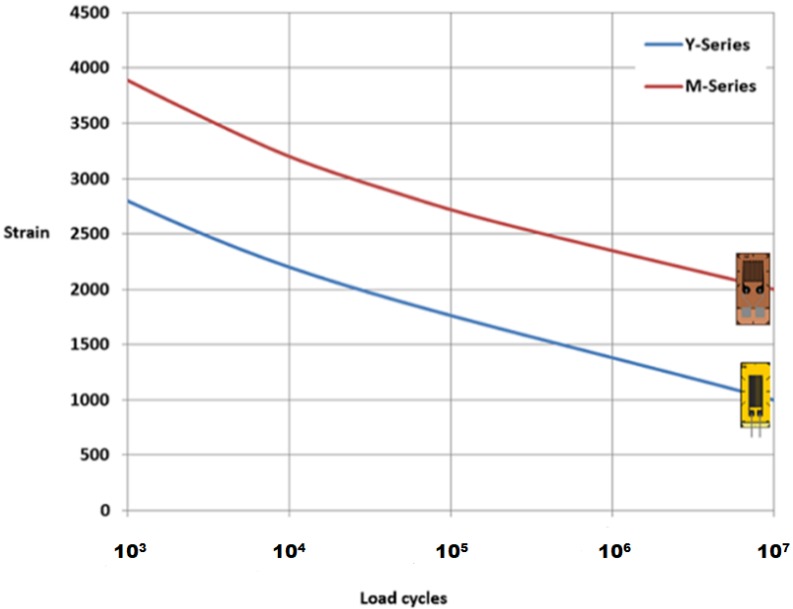
Fatigue life trend plot for commercially supplied standard and fatigue resistant electrical resistance foil gauges [22]. Results came from HBM Company (Got permission from HBM Company).

**Figure 9 sensors-19-03524-f009:**
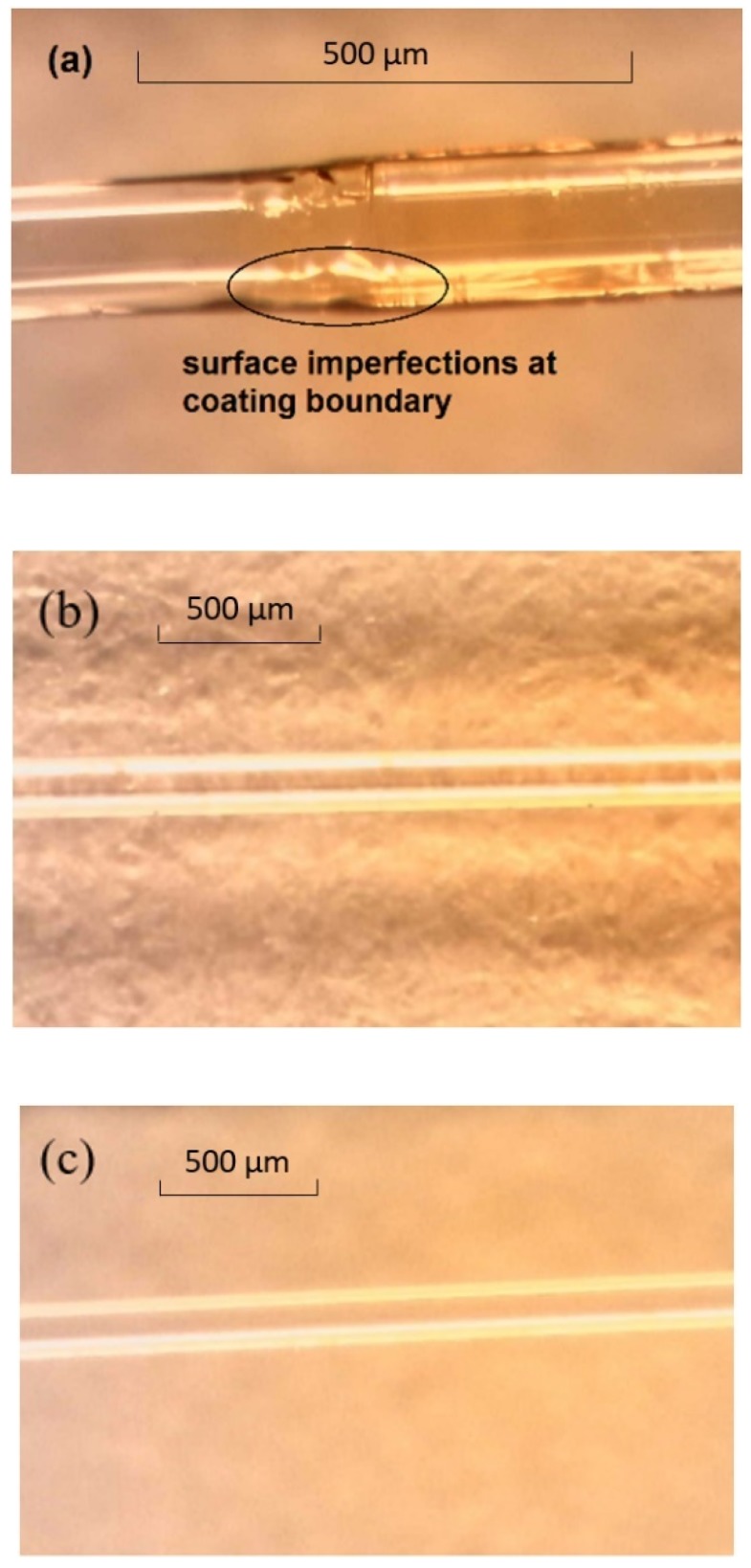
Pre-fatigue microscopic images of fibre surfaces for. (**a**) Stripped and recoated grating. (**b**) Draw tower gratings. (**c**) Trans-jacket gratings. Images (**b**) and (**c**) are at lower magnification to show the entire sensing region.

**Figure 10 sensors-19-03524-f010:**
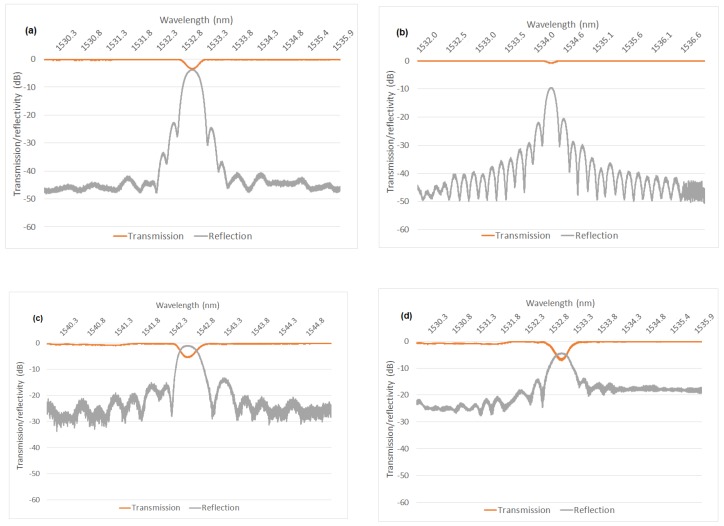
Example transmission and reflection spectra for each type of the FBG sensor. (**a**) Stripped and recoated gratings. (**b**) Draw tower gratings. (**c**) Trans-jacket gratings supplied by research provider. (**d**) Trans-jacket gratings from commercial suppliers.

**Figure 11 sensors-19-03524-f011:**
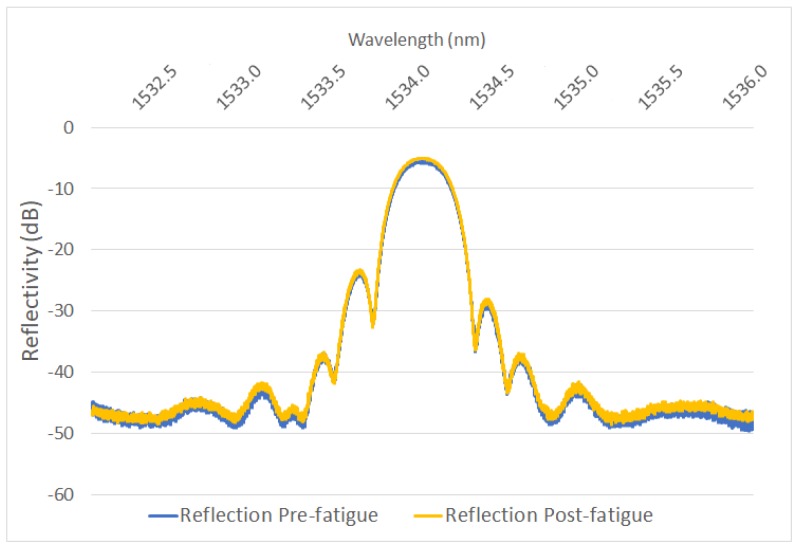
Pre- and post-fatigue reflection spectra comparison show identical reflective wavelength profile for a stripped and recoated FBG which survived the fatigue cycling.

**Table 1 sensors-19-03524-t001:** Finalised test rig parameters.

Parameter	Value	Units
Spring K_1_	4	kN/m
Moving block M_1_	300	grams
Fibre stiffness K_2_ approximation	30	kN/m
Test rig tare mass	3	kg

**Table 2 sensors-19-03524-t002:** FBGs sensor test matrix.

FBG Fabrication Method	Stripped and Recoated FBGs	Draw Tower Gratings (DTGs)	Commercially Sourced Trans-Jacket FBGs	Trans-Jacket FBGs Supplied by a Research Provider
Sample size	15	15	5	10

**Table 3 sensors-19-03524-t003:** Incremental loading sequence for each sensor.

**Peak Strain (µε)**	10,000	15,000	17,000	19,000	21,000	23,000	25,000
**Cycles (Million)**	1	0.5	0.5	0.5	0.5	0.5	0.5

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
