# Peer review of "Fatigue Performance of Type I Fibre Bragg Grating Strain Sensors"

_sensors, 2019, doi:10.3390/s19163524_

Round 1

Reviewer 1 Report

This work has dealt with the fatigue performance of FBG strain sensors. Experimental investigations have been attempted on various types of FBS. While authors interest and efforts are sincerely appreciated, there appears to be several issues to be explained in this work.

1) Table 3: Incremental loading sequence for each sensor. Why the cycles for 10000 microstrain is different with the cycles under the the microstrain?

2) Why the loading frequency was set to 100 Hz?

3) The temperature will increase the FBG fatigue. Could the authors added some experiments or comments on the temperature effcet on the FBG fatigue?

4) The number in Fig.7 should not be half.

5) Microstrain were missed in many sentences.

Author Response

Thank you for your review. We have addressed your specific concerns below.

1) Table 3: Incremental loading sequence for each sensor. Why the cycles for 10000 microstrain   

is different with the cycles under the other microstrain?

Response 1: We initially expected 1 million loading cycles at 10000 microstrain was about fatigue limit of FBGs, which is already a very high loading level. However after a few samples were tested, it was found that every type of sensor survived 10000 microstrain with 1 million cycles. Thus, it was necessary to extend the fatigue cycling regime.  In the later experiments in order to balance the number of cycles with a reasonable testing duration we selected 0.5M cycles of incremental loading until failure or a maximum strain of 25,000 microstrain was reached. This information has been added to the paper.

2) Why the loading frequency was set to 100 Hz?

Response 2: Referring to section 2.1, the test rig has a resonance at about 85 Hz. As the frequency goes up above 85 Hz, the amplification effect of the vibration will drop. However at  85Hz the amplitude of the excursion caused the loading apparatus to become mechanically unstable.  We found in the experiment that around 100 Hz provided the best balance between sufficient vibration amplitude to achieve 25,000 microstrain in the fibre and the testing speed.  This has been articulated more clearly in lines 182-184 of the paper.

3) The temperature will increase the FBG fatigue. Could the authors added some experiments or comments on the temperature effect on the FBG fatigue?

Response 3: Environmental factors such as heat will influence fatigue life. Although typically the fibre protective coating will breakdown and the grating will erase before the temperature reaches a point where the fatigue life of the glass is affected. For these studies the fibres were coated with either ormosil or polyimide (depending on the manufacturer) and the maximum temperature for these coatings before degradation is around 200 - 300 degrees Celsius which is well below the point where the temperature will affect the fatigue life of the glass. The purpose of this study was to understand the effects of the manufacturing process on the fatigue life of the glass, therefore we did not include temperature effects in the study which may be dependent on other things such as the fibre coating type.

4) The number in Fig.7 should not be half.

Response 4: Fixed.

5)      Microstrain were missed in many sentences.

Response 5: Fixed.

Please find the attached revised manuscript,

Reviewer 2 Report

The paper reports on the fatigue performance of FBGs for three different inscription processes.

The report is of great interest to the FBG sensing community.

The paper is well organised and the results are sound. I am happy for the paper to be accepted as is with only one minor comment:

(1)    At numerous locations in the text (e.g. in lines 222, 225, 238, 239, …), the units of measurement are missing. Instead, there are white spaces. It appears that the Greek symbols disappeared during the document conversion.

Author Response

Thank you for your review. We have addressed your specific concerns below. 

1)      At numerous locations in the text (e.g. in lines 222, 225, 238, 239, …), the units of measurement are missing. Instead, there are white spaces. It appears that the Greek symbols disappeared during the document conversion.

Response 1: Fixed. Thanks!

Please find the attached revised manuscript

Reviewer 3 Report

This work reports a study of the mechanical fatigue performance of FBG probes manufactured by different methods, such as the stripping and recoating procedure, the in-drawing writing technique, and the direct grating writing over the jacketed fiber. The subject addressed in this paper is interesting with respect to practical applications of optical fiber sensors in structural health monitoring.

Even though the manuscript has its merits, important information about the materials and the experimental procedure is missing. Moreover, the discussion of the results is too superficial and the authors did not provide a reasonable explanation for the differences between the mechanical characteristics of each FBG. Therefore, I do not recommend this paper for publication in Sensors.

Additional comments about the manuscript:

Sec. 1: the paragraphs between lines 66 and 101 should be moved to the next section for the sake of organization;

Fig. 1 has a low resolution. Please export it with at least 300 dpi. Moreover, the legend is not self-explanatory and must be improved by detailing the parts of the apparatus;

Sec. 2: you must provide the details regarding the fabrication of the FBGs, such as the fiber, the grating periodicity and length, the laser specifications, the applied methods and instruments, etc. since it directly affects the test results. Did you purchase the FBG  from different suppliers?

Line 162: please include the reference for Young’s modulus value;

Sec. 2.1: the information regarding the test rig optimization, as shown in Fig. 3, is superfluous for this manuscript, so it is only necessary to indicate the final values. Moreover, the data shown in Table 1 could be displayed inline in the      previous paragraph;

Sec. 2.2: as the jacket stripping produces damages in the fiber surface, it is necessary to detail the stripping method, the used tools, etc.;

The purpose of Fig. 9 is not clear. Did you test the strain gauges under the same conditions as the FBGs? It would be useful if you plot the data concerning the three types of FBGs in the graph of Fig. 9 for comparing the results;

Fig. 10: please include the scale bars;

Explain (in quantitative terms) the differences between the mechanical properties of the ordinary polymer buffers used in types 1 and 2 FBGs, and the IR transparent jackets applied in type 3 FBGs. In addition, explain why the average failure strain for draw tower gratings is higher than the trans-jacketed gratings;

Sec. 3.1: “…100% of the trans-jacket FBGs made by the research provider survived the fatigue loading sequence…”      Explain the differences between the commercial and the custom-made trans-jacket FBGs and provide the data regarding the loading tests, otherwise, this sentence is not convincing;

Sec. 3.2 and 3.3 are limited to comments about the pictures and the appearance of the FBGs spectra, so there is no discussion of the results.

Author Response

Thank you for your review. We have addressed your specific concerns below. 

Thank you for your review. We have addressed your specific concerns below. 

The gratings were supplied commercially by three different manufacturers and it is difficult to obtain many of the details about their proprietary procedures and processes. We have included additional detail about the fibres and manufacturing processes where available (lines 246-251).

The primary driver behind this work was to compare the fatigue performance of the emerging Type I trans-jacket sensors written with femtosecond pulsed IR lasers to the more common commercially available stripped and re-coated and draw tower gratings for Defence applications.

The trans-jacket inscription process opens up the potential for higher reflectivity, apodisation and more dense or complex array architectures.   There was a difference in performance between the trans-jacket gratings from the commercial and academic suppliers and we have added more detail in the paper to account for this difference. This is not surprising as the progression from the research environment to the commercial environment for this method of manufacturing is very new. 

The conclusions drawn from the paper are that these new class of gratings are expected to provide better performance and lower cost for dense sensor arrays that can be applied in harsh environments as commercial suppliers adopt and refine their processes for trans-jacket inscription. We have added some commentary to the paper to articulate these points more clearly for the reader.

1)      Sec. 1: the paragraphs between lines 66  and 101 should be moved to the next section for the sake of organization;

Response 1: These paragraphs provide the readers with some background information about the different manufacturing processes which are typically used in this field to inscribe the gratings. We feel that this information is better placed in the introduction rather than the experimental section as it helps the reader to understand why the manufacturing process might influence the fatigue performance of the sensor and how this study builds on previously reported work in the area.

2)      Fig. 1 has a low resolution. Please export it with at least 300 dpi. Moreover, the legend is not self-explanatory and must be improved by detailing the parts of the apparatus;

Response 2: Fixed.

3)      Sec. 2: you must provide the details regarding the fabrication of the FBGs, such as the fiber, the grating periodicity and length, the laser specifications, the applied methods and instruments, etc. since it directly affects the test results. Did you purchase the FBG from different suppliers?

Response 3: The gratings were purchased from three different providers. The fibre dimensions, coating thickness, gauge length and centre wavelength were all the same. The gratings were all type I (under the damage threshold of the glass) and the laser type depended on the method of inscription.  Additional details have been included at lines 245-249.

4)      Line 162: please include the reference for Young’s modulus value;

Response 4: Reference included

5)      Sec. 2.1: the information regarding the test rig optimization, as shown in Fig. 3, is superfluous for this manuscript, so it is only necessary to indicate the final values. Moreover, the data shown in Table 1 could be displayed inline in the previous paragraph;

Response 5:  The design and optimisation of the fatigue rig is relevant to the paper for two reasons. Firstly, conventional fatigue testing is typically conducted via servo hydraulic MTS machines which are limited to slow cycle rates. In addition because the dimensions of the fibre are so small, only a small force is required to induce a large strain which means any MTS machine would be operating in the bottom 5% of its capacity which is not recommended for effective load control. Therefore the detailed design of this assembly will be of interest to anyone who would like to apply high-cycle rate fatigue to optical fibres. The optimisation of the rig is a key step and explains to the reader why the cycle rate was limited to 100 KHz and what parameters need to be modified in the design to increase the natural frequency.

6)      Sec. 2.2: as the jacket stripping produces damages in the fiber surface, it is necessary to detail the stripping method, the used tools, etc.;

Response 6:  There are many different ways to remove the fibre coating following three basic classes mechanical, chemical and thermal. For the stripped and re-coated gratings acquired for these studies, the coating was removed by a chemical process. (Heated concentrated sulphuric acid). This detail has been included in the paper.

7)      The purpose of Fig. 9 is not clear. Did you test the strain gauges under the same conditions as the FBGs? It would be useful if you plot the data concerning the three types of FBGs in the      graph of Fig. 9 for comparing the results;

Response 7:  Figure 9 shows the reported fatigue life of electrical resistance strain gauges from HBM who is the biggest commercial supplier (by volume) of electrical strain gauges. They were not tested under the same conditions. The reference for this plot is given in the figure caption and the data is referred to in the body text of the paper as being reported by others. The purpose of including this information is to show the enhanced fatigue performance of all the FBGs (regardless of manufacturing method) over conventional foil gauges.

8)      Fig. 10: please include the scale bars;

Response 8: Scale bars included

9)      Explain (in quantitative terms) the differences between the mechanical properties of the ordinary polymer buffers used in types 1 and 2 FBGs, and the IR transparent jackets applied in type 3 FBGs. In addition, explain why the average failure strain for draw tower gratings is higher than the trans-jacketed gratings;

Response 9: Historically FBGs have typically been written with a UV laser where most of the common fibre coatings are not transparent. This means the coating must either be removed prior to writing the grating or the grating is written in the draw tower prior to coating the fibre. With the advent of stable femtosecond pulsed lasers operating in the infrared region it is now possible to write Type 1 gratings using a phasemask into fibres through standard coatings such as the ones used in this study. There is no difference in the coating just that we can now use laser light of a different wavelength to inscribe through the coating.

10)   Sec. 3.1: “…100% of the trans-jacket FBGs made by the research provider survived the fatigue loading sequence…” Explain the differences between the commercial and the custom-made trans-jacket FBGs and provide the data regarding the loading tests,      otherwise, this sentence is not convincing;

Response 10:

For our custom-made trans-jacket FBGs, an acylindrical lens of short focal length (f=8mm) was used to tightly focus the 800nm femtosecond writing beam in the centre of the fibre. Such tight focusing geometry with reduced optical aberrations allows to reach two orders of magnitude larger intensity around the fibre core compared to the surrounding polymer coating, thus making possible trans-jacket inscription of type-I gratings in the fibre core without damaging the coating. Please find reference 24 in the manuscript. 

11)   Sec. 3.2 and 3.3 are limited to comments about the pictures and the appearance of the FBGs spectra, so there is no discussion of the results.

Response 11

Section 2.3 highlighted both the surface damage evident from the stripping and re-coating process and the lack of surface damage on both the trans-jacket and draw tower grating arrays. The commentary provided linked this experimental evidence to conclusions that the trans-jacket inscription process did not appear to damage the fibre coating. Commentary was also made about the negligible effect of fatigue loading on the coating.

Section 3.3 is an important section and the reflection and transmission spectra provided serve as evidence to support the stated limitations of some of the manufacturing methods. The reduced reflectivity of the draw tower gratings is linked to the speed of the fibre draw process and the limitations which are imposed by this on the exposure time and the ability to apodise the grating profile. The increased reflectivity of the stripped and re-coated and trans-jacket gratings are linked to the processes where the fibre is static and exposure times can be controlled to achieve the required index contrast. The optimised spectral profile of the stripped and re-coated gratings with significantly enhanced side lobe suppression compared to the trans-jacket gratings is linked to the ability to focus the light more effectively in the core without the coating. Some additional text has been included to extend these points.

Please find the attached revised manuscript, the changes are in yellow.

Round 2

Reviewer 3 Report

This work reports a study of the mechanical fatigue performance of FBG probes manufactured by different methods.
Even though the authors managed to improve the paper by addressing most of the reviewers` comments, the manuscript is still unbalanced with excessive details regarding the test rig, whereas important information about the fabrication and the mechanical properties of the FBGs is missing. Moreover, the authors did not improve the discussion of the results.
Therefore, I do not recommend this paper for publication in Sensors, but I strongly advise the authors to submit it to a mechanical engineering-oriented journal.

Author Response

Dear Reviewer 3,

Thank you for reviewing the paper again.

We have made further refinements to the paper in response to your comments. We have removed some of the detail about the fibre fatigue loading design and expanded the discussion section of the paper. In particular we have articulated more directly the relationship between the spectral properties of the gratings and their respective manufacturing processes. We have also included some further details about how the different manufacturing processes might influence the surface quality of the fibre in the region of the sensor.

The gratings investigated as part of this study were supplied commercially by three different manufacturers and it is difficult to obtain many of the subtle details about their proprietary materials and processes. The most significant difference between the three sensor types under investigation from a fatigue performance perspective is the way in which they were physically manufactured. The differences in the manufacturing approaches have been expanded in the introductory section with references provided.

The primary driver behind this work was to compare the fatigue performance of the emerging Type I trans-jacket sensors written with femtosecond pulsed IR lasers to the more common commercially available stripped and re-coated and draw tower gratings for monitoring of high value assets in harsh structural environments.

Trans-jacket FBG sensors have only recently (in the last 2 years) become available commercially and this paper seeks to add to the body of knowledge in the field by firstly providing experimental evidence of fatigue performance and secondly by articulating by the potential benefits of the new manufacturing process for low-cost volume production of complex array architectures.  As such we feel that this paper fits most appropriately in a sensors journal.

Please also find the attached revised paper.